# Does the Addition of Point-of-Care Testing Alter Antibiotic Prescribing Decisions When Patients Present with Acute Sore Throat to Primary Care? A Prospective Test of Change

**DOI:** 10.3390/diagnostics14111104

**Published:** 2024-05-26

**Authors:** Rob Daniels, Esther Miles, Karen Button

**Affiliations:** 1Faculty of Health Care Professions, St Luke’s Campus, University of Exeter, Heavitree Road, Exeter EX1 2LU, UK; 2TASC Primary Care Network, Townsend House Medical Centre, 49 Harepath Road, Seaton EX12 2RY, UK; karenbutton@nhs.net; 3Royal Devon University Hospital, Barrack Road, Exeter EX2 5DW, UK; esther.miles4@nhs.net

**Keywords:** point-of-care-testing, GAS infections, molecular, quality improvement, antimicrobial stewardship

## Abstract

Accurate clinical diagnosis of patients presenting to primary care settings with acute sore throat remains challenging, often resulting in the over-prescribing of antibiotics. Using point-of-care tests (POCTs) to differentiate between respiratory infections is well-accepted, yet evidence on the application within primary care is sparse. We assessed the application of testing patients (*n* = 160) from three family practices with suspected Streptococcal infections using rapid molecular tests (ID NOW Strep A2, Abbott). In addition to comparing clinical evaluation and prescription rates with either usual care or testing, patients and staff completed a questionnaire about their experience of molecular POCT in primary care. The immediate availability of the result was important to patients (100%), and staff (≈90%) stated that molecular testing improved the quality of care. Interestingly, only 22.73% of patients with a Centor score > 2 tested positive for Strep A and, overall, less than 50% of Centor scores 3 and 4 tested positive for Strep A with the ID NOW testing platform. The addition of rapid molecular POCTs to clinical assessment resulted in a 55–65% reduction in immediate and deferred antibiotic prescriptions. The intervention was popular with patients and medical staff but was associated with increased cost and a longer appointment length.

## 1. Introduction

Sore throats remain one of the most common acute presentations in primary care across the UK, accounting for nearly 10% of all GP appointments each year [1]. Although the umbrella term of ‘sore throat’ refers to both pharyngitis and tonsillitis, this study focused primarily on the latter. Differentiating between viral and bacterial causes of tonsillitis by clinical assessment is often difficult, especially in periods with overlapping prevalence. *Streptococcus pyogenes* (Group A Streptococcus; GAS) has become highly adapted to the human host and is the most common pathogenic bacterial cause of tonsillitis [2]. GAS infections can be asymptomatic, presenting as mild conditions such as strep throat or impetigo or develop into severe or life-threatening invasive infections, peri-tonsillar abscess and glomerulonephritis. In primary care, most children present with acute sore throats with or without tonsillar exudate, fever and general malaise, with prevalence highest during the winter months and in the 5–15-year-old age group [3].

The National Institute for Health and Care Excellence (NICE) in the UK recommends prompt treatment with antibiotics in suspected cases of streptococcal infections, with treatment of susceptible close contacts where indicated [4,5]. Currently, the recommended choices are phenoxymethylpenicillin for 5–10 days or Clarithromycin for 5 days in a patient with penicillin allergy [6]. Asymptomatic carriage rates in this age group are around 8% [7], making clinical differentiation of illness rather than carriage important.

Although 70% of cases presenting to primary care are viral, antibiotics are still prescribed for 40% of patients [8]. This overuse of antibiotics has been attributed to the absence of reliable rapid diagnostic tests prior to prescription, with microbiological culture taking up to 72 h before a result is available [9]. Current guidelines, including those from NICE [4], suggest the use of Centor criteria [10] or the FeverPAIN score as decision-making tools to support the clinical assessment. These scores range from 0–4 with points accumulated for specified symptoms or signs: (i) tonsillar exudate, (ii) tender anterior cervical lymphadenopathy or lymphadenitis, (iii) fever (≥38 °C), and (iv) the absence of cough. Individuals with a low total score (less than 2) have a lower probability of streptococcal tonsillitis and therefore antibiotic prescribing may be avoided, while higher scores support either immediate or deferred antibiotic prescription.

Although a meta-analysis [11] showed that Centor criteria may be useful in ruling out GAS if there is a score of 0, scores greater than 1 are associated with variable predictive values of streptococcal infection, with a score of 4 being associated with positive culture in around 55% of cases. Prescribing antibiotics to this cohort of patients following clinical assessment with Centor scoring will therefore result in around 45% of patients who do not have a bacterial infection being treated with antibiotics. Thus, having a combination of clinical scoring and confirmatory testing would strengthen clinical management and reduce unnecessary antibiotic prescriptions.

In comparison to standard microbiological culture, a POC test (either lateral flow or rapid molecular) can result in confirmation in as little as 5–10 min and may be performed on-site during the first patient visit, saving both time and reducing diagnostic uncertainty in the meantime. Collectively, the evidence suggests that the implementation of adjunct point-of-care (POC) testing in addition to scoring systems and clinical judgement [9] may improve the identification of symptomatic streptococcal tonsillitis while reducing unnecessary antibiotic prescriptions. Nardi and colleagues recently demonstrated that the identification of viral aetiology in children using rapid molecular point-of-care testing resulted in a 9-fold reduction in antibiotic prescriptions [12] supporting the potential expansion of POC testing into routine care.

The uptake of POC testing has been slow in primary care, and a study undertaken in Ireland [13] revealed that GPs perceive point-of-care tests as beneficial and would welcome their use in routine clinical practice, citing guidance in decision making as the primary benefit (43%), followed by reduced referral rates (29%) and assistance in diagnosis (13%) [13]. Moreover, during the COVID-19 pandemic, studies in primary care practices in Germany and the United Kingdom using molecular testing reported patient satisfaction in terms of prompt testing and trusted results, reduced further testing and improved workflow and better patient care for high-risk groups [14,15,16].

This quality improvement study sought to investigate the effect of POC testing on antibiotic prescribing in patients presenting to primary care with an acute sore throat. It aimed to determine the financial and time costs associated with POC testing when compared to usual care and assess the feasibility of its implementation. Furthermore, it endeavoured to understand the opinions of patients and staff involved in the testing for a holistic view of POC testing in an NHS setting.

## 2. Materials and Methods

### 2.1. Study Design

Between December 2023 and February 2024, a prospective quality improvement project was designed to evaluate the impact of point-of-care-testing on the management of acute sore throat in ambulatory patients presenting to primary care settings in the towns of Seaton, Colyton and Axminster, in Devon in the United Kingdom. This quality improvement project complied with the General Data Protection Regulation (GDPR) in accordance with the principles of Good Clinical Practice.

### 2.2. Clinical Assessment of Patients

All patients presenting to the General Practice clinics at Townsend House Medical Centre, Seaton and Colyton Medical Practice and Axminster Medical Practice with acute sore throat were assessed according to usual practice. In addition, patients presenting to Townsend House Medical Centre were offered the option of a consultation at Seaton Pharmacy, a local community pharmacy. The pharmacist arm was included to assess the feasibility of pharmacist-led assessment and management of throat infections, prior to the introduction of Pharmacy First, an NHS England initiative to improve access for minor illnesses where pharmacists can treat 7 defined minor illnesses without recourse to a doctor.

All patients were assessed by a clinician. When the decision was then made to consider either immediate or deferred prescription for antibiotics for a clinical diagnosis of streptococcal infection, clinicians were offered the option to follow a new POC pathway or continue with usual care (see Figure 1). When the patient and clinician agreed to follow the POC pathway assessment, the ID NOW Strep A2 test was used as a rapid diagnostic tool. NICE guidance recommends either FeverPain or Centor scoring systems [4,5,6]. In this study, the Centor score was used for patient assessment due to clinician familiarity.

Following clinical assessment, patients with a score of 2, 3 or 4 received a molecular-based POCT Group A streptococcus during the consultation. Nice guidance [4] recommends immediate or deferred prescriptions for patients with a Centor score of 3 or 4; however, during the severe outbreak in the UK between October 2022 and March 2023, this recommendation was amended to a lower threshold for antibiotic prescribing. This amended guideline was removed on 15 February 2023 [17] but high levels of parental anxiety around childhood streptococcal infections remain, which can influence prescribing decisions in the context of shared decision-making discussions between patients (or their parents) and clinicians. Thus, in this study, this protocol was used, which reflects current clinical practice, rather than strict adherence to NICE guidance. Patients who tested positive with molecular POCT were treated according to local protocols [18], while those testing negative received advice on symptom management such as taking paracetamol for fever and pain, adequate intake of fluids or trying medicated lozenges and safety netting advice to seek medical help if symptoms worsen rapidly or significantly, fail to improve after 1 week or the person becomes systematically unwell. An overview of the patient testing synopsis can be viewed in Figure 1.

### 2.3. Point-of-Care Testing Molecular Diagnostics

This project used the Strep A2 test on the molecular Abbott ID NOW^TM^ platform, which is a simple rapid molecular test that is user-friendly and does not require specific microbiology equipment making it ideal for direct POC diagnostics. This molecular isothermal test is based on NEAR (nicking enzyme-combined isothermal amplification) technology, which detects the cepA gene encoding the C5-peptidase with a manufacturer’s specified limit of detection (LOD) between 25 and 147 CFU/mL [9,19]. Following a throat swab, the test results are available after 2 min, with no warm-up time required (Abbott Rapid Diagnostic Ltd., Stockport, UK). Tests were consistently carried out according to the instructions for use for the ID NOW™ and Strep A2 test after all staff involved received training.

### 2.4. Study Population

Patients were recruited during the initial clinical assessment. The inclusion criteria were patients presenting with acute sore throat (less than one week in duration) to one of the three GP practices participating in the study. There were no age or comorbidity restrictions. Patients who fulfilled the inclusion criteria were offered a throat swab for molecular POC testing following initial clinical assessment as described in the section above. Verbal consent from the patient was obtained by the clinician. Patients who declined were managed according to usual clinical care, with the prescribing decision made by the treating clinician.

### 2.5. Data Collection

The nature of the clinical process required minimal demographic information to be recorded, and these were kept on a paper log and stored in a locked filing cabinet in the primary care sites, with access restricted to clinical staff involved in direct patient care. Patient results were transcribed anonymously onto an electronic database using Excel and stored on a secure hard drive at Townsend House Medical Centre.

The following patient data were collected: Name, date of birth, Centor score and POC test result. In addition, the length of consultation time in usual care and POC arms and the time required to process the POC test were documented in a sample of patients. This was performed by the staff at each site responsible for testing.

### 2.6. Questionnaire

In addition to the data on the Centor score and POC test result, patient and clinician satisfaction was also measured using an online survey at the end of the study period, using Online surveys [20]. The questionnaire was based on the expertise of the primary care physicians involved in the study. Patients were asked to answer 5 questions that were rated on a five-point Likert scale (which measured the degree of agreement). All data were collated in an anonymous fashion. All participating members of staff and all patients who received a POC test were invited to complete the questionnaire, which also included items exploring aspects of the usability and feasibility of the POCT.

### 2.7. Statistical Analysis

Results in the POCT group were expressed as a binary outcome: a positive or negative test result. An analysis of subgroups using SPSS was then carried out to determine the relationship between the Centor score and the test result. Appointment duration was measured retrospectively via an analysis of consultation start and finish times in the electronic healthcare records of 40 consecutive patients who had point-of-care testing and 40 who received antibiotics but did not have POC testing. Sample processing time was calculated using a prospective recording of test processing duration by the staff member (pharmacist or nominated staff member in GP practice) using a stopwatch in a sample of consecutive 20 patients.

## 3. Results

A total of 160 patients were included in the analysis and were recruited into the study if the clinical picture reflected a bacterial infection and the Centor score was over 1. The median age of patients enrolled in this study was 17 years. The range was from 2 months old to 86 years old. Furthermore, 36% of patients were male (*n* = 57) and 64% were female (*n* = 103).

### 3.1. Molecular POCT Increases the Diagnostic Accuracy of Clinical Evaluation of Strep A Diagnosis

Figure 2A shows the distribution of Centor scores according to patient age, with significantly more patients with a score of 2 than 4 and a significant correlation (*p* = 0.002) between age and score (r = −0.2970). Following a pharyngeal swab, testing was performed using the ID NOW Strep A2 test and the mean time to process the test was 5.8 min. Of 160 results, a total of 10 tests were invalid, 7 of which were due to operator error in the initial week of the evaluation process. Once retraining had been given, there were three further invalid results, giving an invalid sample rate of 1.9%. Furthermore, 6 patients with a Centor score of 1 were included in error and thus removed from the analysis, giving a total of 144 patients.

Figure 2B shows the comparison of negative and positive ID NOW Strep A2 results according to the Centor Score, and no significant differences were observed between the two groups. Figure 2C depicts the number of patients testing positive or negative for Strep A according to the Centor score and shows that fewer individuals actually tested positive. Table 1 summarises the testing result and clinical picture with the comparison between Centor score grades and the result of the patient’s swab following molecular POCT for Strep A.

Interestingly, only 22.73% of patients presenting a Centor score of 2 tested positive for Strep A. There was a higher correlation of positive results with increasing Centor score assessments. However, overall, less than 50% of Centor scores 3 and 4 were positive for Strep A, with the ID NOW device demonstrating that many cases of throat infection were not due to Strep A. In all of these cases and according to the NICE guidelines, antibiotics would have been prescribed for these patients even though 65.28% of these prescriptions would have been unnecessary.

### 3.2. Longer Consultation Times for POCT Result in Higher Accuracy of Diagnosis

Molecular POCT is not yet a standard practice in primary care offices, and one of the hurdles that needs to be overcome is the concept of longer consultation periods. In this multi-centre evaluation, records were retrospectively reviewed to document the mean length of time required for patients in the normal care pathway that resulted in antibiotics or when a POCT was used prior to prescribing. It was shown that the mean duration of consultation was 17 min for patients presenting with sore throats in the usual care pathway and resulting in an antibiotic prescription (*n* = 40). When a molecular POCT was used, the mean duration of the consultation in the POCT group (including Centor Scoring and the POCT process) was 24 min (*n* = 40), so an average of 7 min longer. In the usual care group, no type of testing such as rapid antigen testing was performed, and evaluations were purely clinical-based. The cohort of individuals in this assessment was a true cross-section of the population since patients ranged from 2 months to the elderly. Although it was not analysed in this study due to a lack of electronic healthcare records to objectively record consultation times, discussions with the pharmacist involved suggest that the addition of point-of-care testing did not lengthen consultations, where a single clinician undertook the clinical assessment, testing and analysis rather than passing the sample to a colleague to analyse, through efficient structuring of consultation tasks. Discussions with clinicians involved also suggested that there is a learning curve and that the duration of consultation was reduced with experience.

### 3.3. Patient Responses Following the Availability of Immediate Test Results

Among the POCT cohort, 21 out of 144 (14.6) patients consented to engage in the survey. Scoring was formatted using a Likert scale (Figure 3) with dark blue and orange bars denoting high agreement with grey and yellow indicating disagreement. Light blue bars showed undecided opinions or no answer. Overall, patients considered having the test performed on-site excellent (100%) and 88% would like the test to be available in the future. Nearly 90% of patients considered it reassuring to learn the correct diagnosis during the consultation period and over 80% considered that patient care and management were improved. Some statements from the patients included “Great service, great to be able to have test done there and then speed, accuracy, reassurance and quick correct treatment as result”, “Excellent service. Quick results resulting in no need for antibiotics which is always the best outcome if possible” and “As my daughter has strep throat and glandular fever last May, it was important that she was tested for strep throat, otherwise she could of fallen ill again. It was do quick and easy to go the swab test at the GP surgery, we waited 5 min for the results. And it put our minds at rest”.

### 3.4. Staff Survey

At the end of the study, a total of 29 staff members from the four primary care sites responded to a survey to share their opinions about the implementation of molecular POCT per se. Respondents included 15 doctors (53.6%), 3 paramedics (10.8%), 4 nurses (14.3%), 3 administration staff (10.8%), 3 practice managers (10.8%) and 1 pharmacist (3.6%). Of these individuals, 82.8% had been directly involved in the POC testing process, providing viewpoints across the entire practice staff. The Paramedics who responded are employed in primary care rather than emergency care roles under the NHS England Primary Care Network Additional Roles Reimbursement Scheme. The collated results of the staff survey are shown below in Figure 4, and the results are presented in the Likert format with dark blue and orange denoting agreement whereas yellow and light blue show disagreement. Grey and green show undecided and not applicable results, respectively. In total, 72% of staff members would like to continue using molecular POCT in the future and most found it easy to use. Over 90% stated that it improved treatment options, which was also reflected in the positive outcome for reducing antibiotics, further microbiological testing and consultations for sore throats. Strong disagreement was noted in administration aspects. Some overall individual comments from the different staff members included “This equipment has changed my practice. Much better than centor or fever pain”, “It increases administrative time but decreases clinical time overall”, “…this QI project has shown that by doing the tests antibiotics have not been prescribed when they otherwise would have been and the patient is reassured. It also means that contact tracing and antimicrobial use is targeted”.

## 4. Discussion

According to the recommendations outlined in the NICE guideline, antibiotic prescription is advised for patients with Centor Scores of 3 or 4. Within the purview of our investigation, a total of 78 patients (53 patients scored 3 and 25 patients scored 4) were identified with a Centor Score of 3 or 4, indicating the need for antibiotic treatment. However, our quality improvement project revealed that 43 out of these 78 patients tested negative for Strep A following molecular POC testing. This finding suggests that 55% of the study cohort assessed as having high Centor Scores did not require an antibiotic prescription (immediate or delayed). This demonstrates that the utilization of molecular POC testing effectively reduces unnecessary antibiotic prescribing while continuing to target patients with a confirmed streptococcal throat infection. Interestingly, a primary care study using lateral flow tests also noted a trend in reduced antibiotic prescriptions in children presenting a sore throat score over 3 and noted that prescriptions were provided due to a lack of trust in the lateral test result [21]. In this study, both patients and doctors trusted the outcome of the rapid molecular test, which corresponds to opinions and attitudes observed in earlier studies carried out in primary care sectors [14,15,16].

Although NICE guidelines recommend the utilization of scoring systems to determine antibiotic prescriptions, clinicians also need to manage patients’ (or their parents’) ideas, concerns and expectations around prescribing, particularly in the context of recent increases in the incidence of streptococcal infections [22]. A recent study conducted in England revealed a notable rise in patient demand for antibiotics, which increased from 65% to 84% between 2010 and 2017. Through corresponding analyses, a positive correlation was established between increasing demands on GPs and their propensity to prescribe broad-spectrum antibiotics [23]. Point-of-care testing in combination with clinical decision making has the potential to significantly improve antimicrobial stewardship by reassuring patients that antibiotics are not required, as shown in the patient survey carried out as part of this study.

The need for a more systematic and accurate process for the diagnosis and management of suspected streptococcal throat infection was highlighted by the outbreak of invasive group A streptococcal infection in the UK in the autumn and winter of 2022/2023 [22]. During that season, there were significant increases in children presenting with not only non-invasive group A strep infections (tonsillitis and scarlet fever) but invasive infections too with several reported deaths. According to the UK Health and Security Agency, a total of 58,972 cases of scarlet fever were reported from week 37 to week 24 of that season (2022 to 2023) in England, with a pre-Christmas peak of 10,069 cases in week 49. This compares with an average of 12,906 cases (range 1296 to 28,303) for this same time period (weeks 37 to 24) in the previous 5 years [24]. This increase in primary care notifications for scarlet fever was accompanied by a similar increase in GP consultations for sore throat and high levels of anxiety.

As a result of the significant increases seen in the incidence of tonsillitis, scarlet fever and invasive group A streptococcal infection combined with national guidance to reduce the threshold for prescribing antibiotics [25] and antibiotic prescriptions in primary care increased significantly, with parental anxiety playing a significant role. At the peak of the above-mentioned outbreak, prescribing of phenoxymethylpenicillin was 6.1 times higher than in the equivalent time period during previous years [26] and serious shortage protocols were required for several antibiotics to allow pharmacists to make substitutions where prescribed antibiotics were unavailable [27]. Suspected group A streptococcal infections can result in much higher prescribing rates than other bacterial infections due to the need to treat close contacts, further highlighting the risks to supply chains and the need for accurate point-of-care testing.

POCT is defined as a diagnostic test, performed during the first patient consultation with test results available within 15 min. Using an oral-based swab, the ID NOW Strep A2 test can provide a positive molecular assessment within 2 min and has a documented sensitivity of 98.5% and specificity of 93.4% [19]. As well as Strep A, the platform can detect viral infections (influenza A&B, RSV and SARS-CoV-2), making it a potentially useful adjunct to standard clinical assessments, and has been shown in hospital settings to improve clinical management of patients, reduce antibiotic prescription rates and reduce unnecessary secondary testing [28,29,30,31]. Previously, this group and others have assessed the practicality of implementing rapid molecular POCT for SARS-CoV-2 in primary care settings [16]. During the evaluation period here, it was determined that POCT improved the clinical diagnosis of Strep A when compared to usual care (Figure 2).

The results presented here suggest that the addition of point-of-care testing to usual clinical assessment significantly reduces antibiotic prescribing and is popular with patients and medical staff but is associated with additional costs and consultation time. In association, earlier studies with the ID NOW testing platform have shown reduced antibiotic prescriptions (up to 9-fold) and use (33%) [12,23]. It was also shown that 30% of physicians changed the care plan for the patient following correct diagnosis [32]. Although the additional cost of the point-of-care testing equipment is £20 per tested patient, this is offset by lower costs for microbiology throat swabs (NHS tariff cost £12 [33]), a potentially lower rate of reattendance for patients managed with reassurance and symptom control at first contact and reduced antibiotic prescribing costs. Indeed, Nardi and colleagues showed that 50% fewer costs were incurred for antibiotics following the implementation of molecular POCT [12].

The 65% reduction in immediate or deferred prescriptions seen in the project reflects the current clinical practice in the study setting, where a Centor score of 2 will usually result in a deferred prescription after a discussion with patients, given high residual anxiety around group A streptococcal infections. Applying the reinstated NICE guidance [17] with a higher threshold of 3 or 4 for antibiotic prescribing would result in a 55% reduction in antibiotic prescribing.

These reductions in antimicrobial prescribing would be very beneficial during future streptococcal outbreaks as seen in 2022/3 when many pharmacies were unable to supply antibiotics due to unprecedented numbers of prescriptions for possible cases of streptococcal infection and their contacts [30]. Better targeting of antimicrobials to patients most in need is also a key component of attempts to reduce antimicrobial resistance, and this study suggests that point-of-care testing may have a significant role to play. In England in 2022, 2,896,095 prescriptions were issued for phenoxymethylpenicillin [34], the majority of which will have been for sore throat. Assuming that these were all prescribed according to NICE guidance, if the results in the project were replicated, the addition of point-of-care testing to usual care would be expected to reduce this significantly. Additional benefits would also be expected from the rapid assessment of cases in terms of limiting outbreaks through early identification and isolation of cases and reduced time off work and school for patients and their caregivers, although this was not assessed in this study.

The successful inclusion of a community pharmacy in this project also suggests that community pharmacists can effectively provide extra capacity in managing acute respiratory infections, either as a regular service or as surge capacity in epidemics as seen in the UK in winter 2022/3. The recent advent of the Pharmacy First scheme [35] provides a dedicated minor illness service that could be expanded to incorporate this service. Comparisons with previous years’ data are difficult as the incidence of streptococcal throat infections varies greatly year by year, making a comparison of antibiotic prescribing rates between winter 2023/4 and 2022/3 unreliable. Although it would be possible to compare prescribing rates with a neighbouring Primary Care Network with similar demographics, the relatively small number of patients (160 in this study) will make statistically significant comparisons difficult to obtain. Similarly, it has not been possible to identify changes in terms of re-consultation or use of secondary care services for the same reasons.

This project will continue beyond the end of February 2024 to collect more data, allowing the collection of a larger dataset, facilitating a comparison of patients managed with or without POC testing and also allowing a subgroup analysis of patients with different Centor scores to determine the correlation between Centor score and POC result. The data presented here do suggest that the POC test result is in line with Centor predictions, but the sample size for subgroups is small. In addition, we intend to look at the relationship between Centor score, Fever Pain score [36] and POC test results in the future.

Although this project took place in three GP practices and a community pharmacy, the population of Seaton and Axminster has a relatively low proportion of deprived and ethnic minority patients and is semi-rural with the nearest acute hospital 23 miles away, with relatively poor transport links. As a result, locality-based interventions may be more effective due to the limited choice of accessible alternative providers. Whether these results would be seen in a more diverse metropolitan population is unclear, but the inclusion of community pharmacies does suggest that in urban areas, the additional capacity they might offer may be a valuable adjunct to current services and reduce demand on the emergency department and out-of-hours services. Further studies in other settings would shed more light on this, and the introduction of Pharmacy First will offer the opportunity to study this in more detail.

The limitations of the project described here include the lack of a control group. This was a quality improvement project rather than a clinical trial and used an intervention at the point where a decision had been made by the treating clinician to use either a deferred or immediate prescription, rather than patients being allocated to control or intervention at the point of entry into the primary care assessment pathway. Consultation durations for both POC and usual care pathways were longer than expected, which may reflect the fact that the majority of these consultations in this study were undertaken by nurse practitioners or primary care paramedics, rather than doctors, who tend to have shorter appointments. Further work assessing workflow aspects of this care pathway would be useful to further explore the economic impact of POC testing.

## 5. Conclusions

Streptococcal throat infections cause significant morbidity and mortality and are responsible for large numbers of antibiotic prescriptions in primary care while existing clinical judgement and diagnostic support tools have poor sensitivity and specificity. The results of this project demonstrate that point-of-care testing has the potential to significantly improve the accuracy of diagnosis at the first consultation, and in our study, antibiotic prescriptions were reduced by 55% following the addition of POC testing to usual care with clinical examination, history taking and Centor scoring. In addition, the ease of use in non-medical settings offers the potential to significantly expand the provision of community-based acute respiratory infection services. The use of this technology in routine practice is popular with patients and staff but does increase consultation times by around 7 min.

## Figures and Tables

**Figure 1 diagnostics-14-01104-f001:**
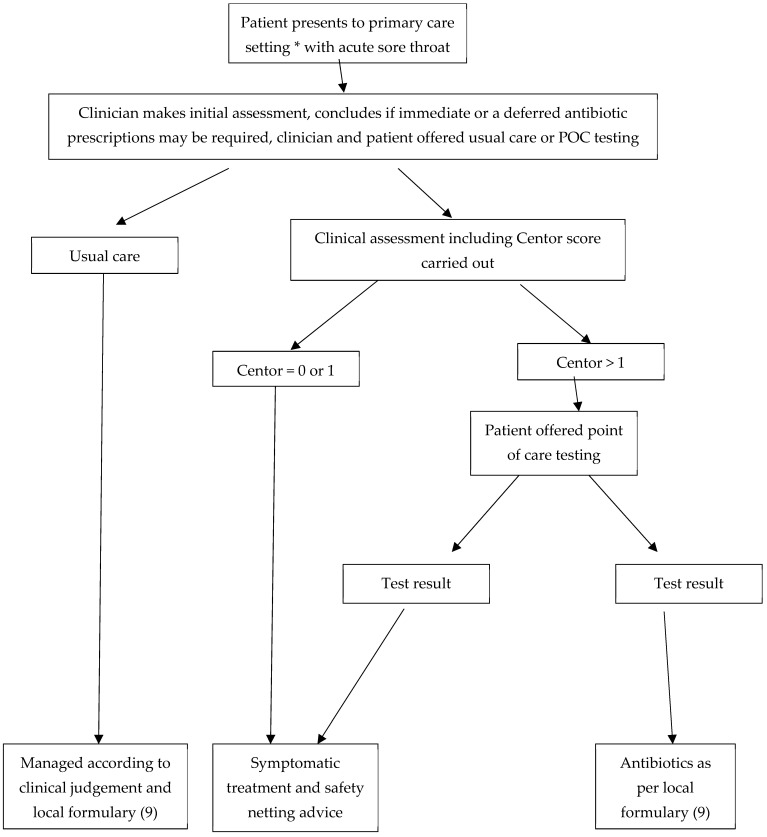
Patient flow and clinical management. * Patients were offered a choice of initial consultation at either GP surgery or high-street pharmacy, according to patient choice.

**Figure 2 diagnostics-14-01104-f002:**
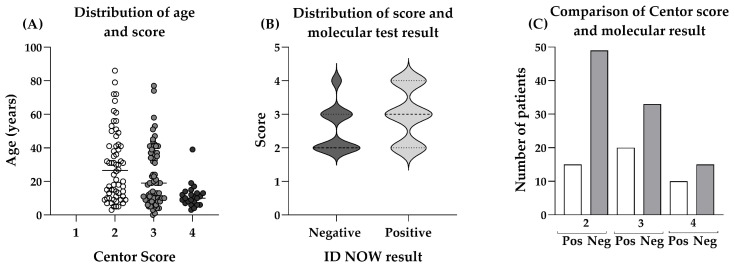
Comparisons of Centor score, age and molecular test result. (**A**) Distribution of Centor score according to age (2 months to 86 years). Statistical significance using *t*-tests between the individual groups was between scores 2 and 3 (*p* ≤ 0.01), between scores 2 and 4 (*p* ≤ 0.0001) and between scores 3 and 4 (*p* ≤ 0.001). (**B**) Comparison of positive and negative molecular Strep A results according to the Centor Score. (**C**) Number of patients presenting with either Centor scores of 2, 3 or 4 and their corresponding positive and negative molecular test result for a Strep A infection.

**Figure 3 diagnostics-14-01104-f003:**
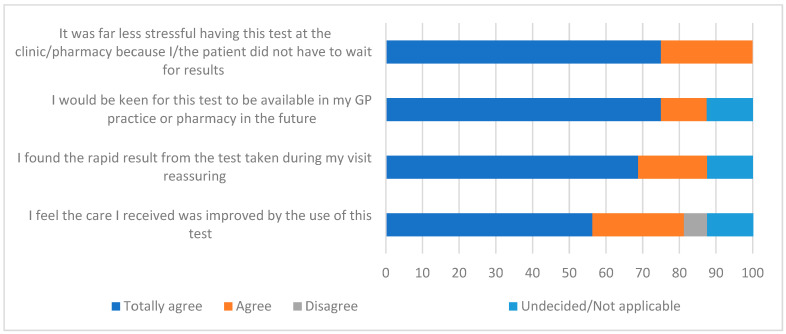
Rating of available molecular point-of-care testing for Strep A in primary care practices. Patients who were offered and agreed to have a molecular test for Strep A at the practice office further participated in an online anonymous survey. Following a Likert format, patients rated their responses from full agreement to full disagreement or not applicable.

**Figure 4 diagnostics-14-01104-f004:**
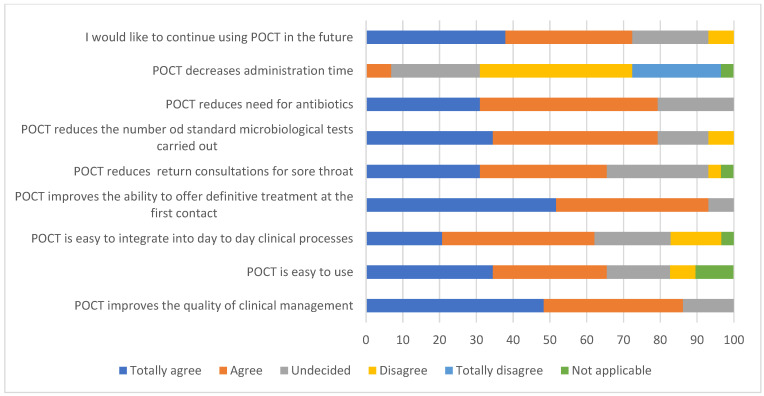
Perceptions of primary practice staff members regarding molecular POC testing for cases involving sore throat. From the participating four practices in Devon, all staff members completed an online questionnaire, which is presented using a Likert scale ranging from agreement to disagreement.

**Table 1 diagnostics-14-01104-t001:** Comparison of Centor score grading and outcome of Strep A testing using the ID NOW^TM^ platform. Following a clinical assessment for sore throat symptoms, oral swabs from 144 patients underwent further molecular diagnosis. Data show the number of positive POCT results compared to those testing negative and the overall percentage of positive individuals per score.

Centor Score	POCT Results: Positive	Negative	% Positive (95% Confidence Interval)
2	15	51	22.73 (14.18 to 34.28)
3	20	33	37.74 (25.91 to 51.22)
4	15	10	60.00 (40.70 to 76.64)
3 or 4	35	43	44.87 (34.33 to 55.89)
Total	50	94	34.72 (27.42 to 42.81)

## Data Availability

The data used in this study are not publicly available due to data privacy restrictions on other health-related issues of the participating individuals.

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
