# Peer review of "Does the Addition of Point-of-Care Testing Alter Antibiotic Prescribing Decisions When Patients Present with Acute Sore Throat to Primary Care? A Prospective Test of Change"

_diagnostics, 2024, doi:10.3390/diagnostics14111104_

Round 1

Reviewer 1 Report

Comments and Suggestions for Authors

Thank you for conducting this research. I have few comments to be addressed:

1- The title does not accurately reflect the relationship between the POCTs and antibiotic prescribing. The manuscript doesn't include any surveillance work related to antibiotic prescribing or consumption. I suggest modifying the title accordingly.

2- The conclusion must be more accurate also to suggest potential effect on antibiotic prescribing.

3- The first paragraph in results section (page 5 lines 214-224) belongs more to the discussion section.

4- I suggest not to elaborate regarding costs in the discussion since this is not lying within the objective of the study and deserves a different research design.

Author Response

Many thanks for your helpful comments. I have made the following changes:

1- The title does not accurately reflect the relationship between the POCTs and antibiotic prescribing. The manuscript doesn't include any surveillance work related to antibiotic prescribing or consumption. I suggest modifying the title accordingly. I have amended to ‘Does the Addition of Point of Care Testing Alter Antibiotic Prescribing Decisions When Patients Present with Acute Sore Throat to Primary Care? A Prospective Test of Change'

2- The conclusion must be more accurate also to suggest potential effect on antibiotic prescribing. I have amended to 
‘. Conclusions
Streptococcal throat infections cause significant morbidity and mortality and are responsible for large numbers of antibiotic prescriptions in primary care, while existing clinical judgement and diagnostic support tools have poor sensitivity and specificity. The results of this project demonstrate that point of care testing has the potential to significantly improve the accuracy of diagnosis at the first consultation and in our study antibiotic prescriptions were reduced by 55% following the addition of POC testing to usual care with clinical examination, history taking and Centor scoring. In addition, the ease of use in non-medical settings offers potential to significantly expand the provision of community based acute respiratory infection services. The use of this technology in routine practice is popular with patients and staff but does increase consultation times by around 7 minutes.”

3- The first paragraph in results section (page 5 lines 214-224) belongs more to the discussion section. I have moved this section as recommended. Thank you. 

4- I suggest not to elaborate regarding costs in the discussion since this is not lying within the objective of the study and deserves a different research design. I have removed this section of the discussion

Reviewer 2 Report

Comments and Suggestions for Authors

The manuscript is well-written, and important topic for research, here are points for further improvements

1- introduction is very long try to reduce it to

2- in the method, elaborate more on Centor score. what should each score do based on NICE guideline

3- In the medthod, study population, you should not include results here, move to the results section, 

4- method, you need to indicate what software did you use to do statistical analysis

5- the first 2 paragraphs in the results are massive and not easy to read. instead of heavy text, you should have Table 1 and move some of your results there

6- list of pricing in the discussion section is not appropriate, please reformat

7-limitation should be last paragraph of discussion

Comments on the Quality of English Language

English is good 

Author Response

Thank you for your comments. All changes have been made as suggested. 

Reviewer 3 Report

Comments and Suggestions for Authors

I think that the manuscript entitled “Does Point of care testing reduce antibiotic prescribing in patients presenting with acute sore throat to primary care? A prospective test of change.” is in principle suited for a publication in Diagnostics, Special Issue “Point-of-Care Testing for Infectious Diseases, 2nd Edition”. The content of the study holds significant importance within the realm of primary care and antibiotic stewardship. With the rise of antimicrobial resistance and the need for targeted antibiotic prescribing, the integration of POCT for streptococcal infections offers a promising solution. Overall, the presentation of the study is clear, organized, and thorough. The utilization of the Abbott ID NOWTM platform for molecular POCT is well-justified, supported by previous research, and adheres to standard guidelines. Additionally, the inclusion of patient and staff feedback adds a human element to the research, enhancing its appeal to readers. However, I have some comments.

Comments:

Figure 1. The scheme in Figure 1 should be presented on one page to avoid splitting the diagram into two pages and complicating its readability.

Lines 175-176. "36% of patients were male (n=57), 37% female (n=103)." If the total number of patients included in the analysis is 160, then correct the percentage of females – 64%.

Figure 2. What do the "?" signs in Figure 2 (A) mean?

Lines 299-324. It is not entirely clear whether all 29 staff members had direct experience conducting the molecular POCT procedure to answer questions such as "POCT is easy to use," "I would like to continue using POCT in the future."

Lines 468-472.  It is noted here that Figures 1-4, as well as Table 1, belong to supplementary materials. Although the titles of these figures and the table fully replicate the titles of those included in the main text of the manuscript. Please check if it is necessary to add Supplementary materials. I believe that only those figures and tables that are not repeated in the main text of the manuscript should be included in the supplementary materials.

Author Response

Thank you. all changes made as suggested

Round 2

Reviewer 3 Report

Comments and Suggestions for Authors

The authors have made the necessary changes to the manuscript. I believe the manuscript can be published after minor text editing. Specifically, the font of Table 1 and the captions for Figure 2 need to be adjusted to match the font used in the main text.

Author Response

Thank you for your feedback. I have made the changes as suggested.  Many thanks

Dr Rob Daniels